# The Comprehensive Analysis of Specific Proteins as Novel Biomarkers Involved in the Diagnosis and Progression of Gastric Cancer

**DOI:** 10.3390/ijms24108833

**Published:** 2023-05-16

**Authors:** Elżbieta Pawluczuk, Marta Łukaszewicz-Zając, Barbara Mroczko

**Affiliations:** 1Department of Neurodegeneration Diagnostics, Medical University of Bialystok, 15-269 Bialystok, Poland; elzbieta.pawluczuk16@wp.pl (E.P.); mroczko@umb.edu.pl (B.M.); 2Department of Biochemical Diagnostics, Medical University of Bialystok, Waszyngtona 15a, 15-269 Bialystok, Poland

**Keywords:** biomarkers, gastric cancer, DNA-based markers, chemokines, MMPs, TIMPs, novel biomarker

## Abstract

Gastric cancer (GC) cases are predicted to rise by 2040 to approximately 1.8 million cases, while GC-caused deaths to 1.3 million yearly worldwide. To change this prognosis, there is a need to improve the diagnosis of GC patients because this deadly malignancy is usually detected at an advanced stage. Therefore, new biomarkers of early GC are sorely needed. In the present paper, we summarized and referred to a number of original pieces of research concerning the clinical significance of specific proteins as potential biomarkers for GC in comparison to well-established tumor markers for this malignancy. It has been proved that selected chemokines and their specific receptors, vascular endothelial growth factor (VEGF) and epidermal growth factor receptor (EGFR), specific proteins such as interleukin 6 (IL-6) and C-reactive protein (CRP), matrix metalloproteinases (MMPs) and their tissue inhibitors (TIMPs), a disintegrin and metalloproteinase with thrombospondin motifs (ADAMTS), as well as DNA- and RNA-based biomarkers, and c-MET (tyrosine-protein kinase Met) play a role in the pathogenesis of GC. Based on the recent scientific literature, our review indicates that presented specific proteins are potential biomarkers in the diagnosis and progression of GC as well as might be used as prognostic factors of GC patients’ survival.

## 1. Gastric Cancer

According to the newest data from WHO, published in 2020, gastric cancer (GC) remains the fifth most common cancer worldwide. Moreover, it is the third most often cause of cancer-related death in the world [1]. The Lauren classification divides gastric adenocarcinoma into two histopathological types: diffuse and intestinal type [2]. The diffuse type is more often observed among young women and subjects with cancer-positive histories. The intestinal type of GC is mostly connected with chronic atrophic gastritis, which can develop into intestinal metaplasia via dysplasia and then can transform into carcinoma in situ. This type occurs more often in older patients, men and people from high-risk countries [3,4]. The risk factors of GC are *Helicobacter pylori* (*H. pylori*) and Epstein-Barr virus (EBV) infections, as well as chronic inflammation process, alcohol consumption, smoking, a diet high in salt, obesity, and also a lack of fruit and vegetables. Risk factors are connected with working conditions such as chemical exposure and working with metal, wood and rubber [5]. The stomach has several anatomical parts: cardia, fundus, body, antrum and pylorus. There are also risk factors connected with adenocarcinoma arising from the cardia (cardia GC) and from other parts of the stomach (non-cardia GC). Cardia and non-cardia GC are both correlated with older age, male sex, tobacco smoking, race, past cancer family history, low intake of fiber, low physical activity and radiation. Development of cardia GC is associated with obesity and gastroesophageal reflux disease (GERD), while non-cardia GC is correlated with *H. pylori* infection, low socioeconomic status, high intake of salty and smoked food and low consumption of fruits and vegetables [6]. Despite the knowledge about risk factors, the prognosis of GC is still poor, and the five-year survival rate is lower than 30% [7].

The diagnostic methods of GC are mainly invasive, including gastroscopy with a biopsy. In clinical practice, there is also a supportive role of imaging methods like computed tomography (CT) and magnetic resonance imaging (MRI), positron emission tomography (PET) and endoscopic ultrasound scanning (EUS) [8]. In addition, laboratory tests are also important tools in the diagnostic process of patients with this malignancy. Well-established tumor markers useful in the routine diagnosis of GC patients are carbohydrate antigen 72-4 (CA 72-4), carcinoembryonic antigen (CEA) and carbohydrate antigen 19-9 (CA 19-9). The concentrations of these proteins may also be elevated in patients with other nonmalignant diseases; thus, their diagnostic specificity and sensitivity are not significantly high enough to be used in screening tests that can accurately diagnose early-stage GC. Therefore, there is still a need to find new potential biomarkers for the early detection of this malignancy. In this review, we present a number of research papers, concerning the significance of potential biomarkers as novel perspectives in the diagnostic process of GC, in comparison to classical tumor markers for this common malignancy (Figure 1, Table 1) [9,10,11,12,13,14]. 

## 2. Biomarkers of GC Applied in Clinical Practice

The diagnostic and prognostic significance of biochemical classical tumor markers in routine practice has been demonstrated. Cancer antigen—CA 72-4 was proved to be the first-line tumor marker with the highest diagnostic sensitivity and specificity for GC. This protein was detected in the 1980s as an antigen reactive to murine antibodies produced by mice immunized with human metastatic breast cancer. Its level is elevated among patients with gastric cancer, colorectal cancer, liver cancer, pancreatic cancer, breast cancer and ovarian cancer [11,12,13,29]. It is considered to be one of the most specific and sensitive GC biomarkers. In the study by Chen et al., the diagnostic sensitivity of CA 72-4 in GC was 32.2%, and the specificity was 97.1% [11,12,13,29]. 

The second line tumor marker in the diagnosis of GC patients is CEA. This protein was first isolated by Gold and Freedman in 1965, and it became a first-line biomarker for colorectal cancer [30]. However, it is used mainly as a marker for digestive system tumors, including GC. The diagnostic specificity and sensitivity of only CEA concentration measurements are not high enough to establish a recognition of this cancer. Thus, in routine practice, the assessment of serum CEA levels is used to monitor the treatment process in colorectal cancer patients and to identify recurrences after surgical resection of a tumor. Serum levels of CEA are elevated in the blood of patients with pancreatic cancer, gastric cancer, breast cancer, lung cancer, cervical cancer, bladder cancer, lymphomas, as well as patients with nonmalignant diseases such as gastric ulcer and duodenal ulcer, colitis ulcerosa, cirrhosis and chronic pancreatitis [13,14,31,32,33,34]. In the study performed by Hao et al., elevated serum level of CEA was identified in 49 diseases [31]. The highest CEA concentrations in descending order had patients with lung fibrosis, pancreatic cancer, uremia, chronic obstructive pulmonary disease, colon cancer, Alzheimer’s disease, rectum cancer, and lung cancer [31]. In the study by Chen et al., the diagnostic sensitivity of CEA in GC diagnosis was 28.7%, and the diagnostic specificity was 96.2% [11].

Cancer antigen—CA 19-9 is also useful in the diagnosis of GC. It is a high-molecular-weight mucin that takes part in the adhesion of endothelial and cancer cells. This protein was detected by mice immunization by a human colorectal cell line. Some researchers indicated the usefulness of CA 19-9 measurement as a marker for gallbladder cancer, bile duct cancer, pancreatic cancer and colorectal cancer [13,35,36,37]. In the study by Chen et al., it was established that the serum level of CA 19-9 was higher in GC patients than in the control group and in benign lesions [11]. The diagnostic sensitivity of serum CA 19-9 measurement in GC patients was 26.4%, and the specificity was 99%. Moreover, elevated serum CA 19-9 concentration was proved to be an independent prognosis predictor in GC patients with metastasis or recurrent cancer [38,39,40].

Some clinical investigations have indicated that human epidermal growth factor receptor 2 (HER2) might also be an important biomarker of GC, especially among patients with an advanced stage. The evaluation of HER2 levels is used to select subgroups for treatment with trastuzumab, and its level is assessed by immunohistochemical technique and FISH methods (fluorescence in situ hybridization) [41,42]. The introduction of trastuzumab led to HER2-positive GC diagnostics. In addition, the association between HER2 expression and treatment of colon cancer, bladder cancer and biliary cancer patients has also been evaluated [9,10]. It is expected that more anti-HER2 medications will be introduced into clinical practice in the treatment process of HER2-positive cancers, including GC [9,10]. 

## 3. Perspective for Novel Biomarkers in GC

### 3.1. Chemokines and Their Specific Receptors

Chemokines are small proteins from 8 to 14 kDa secreted by leukocytes, endothelium cells, fibroblasts, monocytes and tumor microenvironment cells. These cytokines are involved in various physiological processes such as cell migration, adhesion and activation of leukocytes as well as inflammation and immunological response. Moreover, chemokines may also regulate pathological processes, including autoimmunological diseases as well as tumor growth via the promotion of the proliferation of malignant cells and neoangiogenesis [43,44,45].

The biological functions of chemokines are activated via their specific, seven-transmembrane G-coupled receptors [45]. Some investigators suggest that selected chemokines and their specific receptors are connected with GC pathogenesis and may be used as potential GC biomarkers. Our previous findings indicated the significance of various chemokines and their specific receptors as novel biomarkers of gastrointestinal cancers, including GC. Our results suggest the role of serum C-X-C motif chemokine ligand 8 (CXCL8) and its specific receptor (CXCR2) as a promising candidate for a biomarker in the GC diagnosis [15]. We assessed that the serum level of CXCL8 and CXCR2 was significantly higher among GC patients than in the healthy control group. The diagnostic sensitivity of CXCL8 was higher than the well-known tumor marker, CEA, and higher than the specific receptor for CXCL8 (CXCR2). Combined analysis of CXCL8 and CA19-9 increased the diagnostic sensitivity up to 89%. Positive predictive values (PPV) for CXCL8 and CXCR2 were higher than for the classical tumor marker CA19-9. In addition, we proved the role of CXCL8 as a potential GC biomarker, especially in combined measurement with classical tumor markers [15]. In the study of Wei et al., the expression of other chemokines from C-X-C family chemokines—CXCL13—was analyzed. High CXCL13 expression was associated with a larger tumor diameter and a shorter overall survival rate. In addition, low CXCL13 expression was connected with longer survival, especially in the group of patients who received adjuvant chemotherapy. Authors conclude that CXCL13 expression may be a predictive biomarker for postoperative adjuvant chemotherapy benefit in GC patients [16].

A study by Chen et al. revealed that CXCR3 expression was higher in GC tissues than in precancerous tissues. Moreover, overexpression of CXCR3 correlated with decreased alternatively activated macrophage infiltration (M2 macrophages), and patients with decreased infiltration had also a better overall survival rate [17]. The lower expression of CXCR3 correlated with worse differentiation, more advanced GC stage and higher depth of invasion. They assumed that CXCR3 expression might be used as an independent prognostic parameter for the overall survival of GC patients [17]. 

### 3.2. Vascular Endothelial Growth Factor (VEGF) and Epidermal Growth Factor Receptor (EGFR)

Vascular endothelial growth factor (VEGF) or vascular permeability factor (VPF) is a growth factor that stimulates the formation of blood vessels. It also increases vascular permeability, promotes cell migration and regulates the normal and pathological angiogenic processes [46]. VEGF binds to its receptor, VEGFR, and promotes vascularization, vasodilatation and vascular growth [47]. Epidermal growth factor receptor (EGFR) belongs to the tyrosine kinase receptors family. It is a key regulator in cell proliferation, differentiation, survival and cancer development [48]. In the Lieto et al. study, VEGF and EGFR levels were assessed to evaluate whether these molecules might be used as novel biomarkers of GC. VEGF expression was positive in 48% of assessed samples and EGFR in 44% of samples. Curatively treated GC patients had lower VEGF and EGFR expression than noncurative-treated GC cases. In addition, elevated VEGF and EGFR expression was associated with nodal invasion and tumor progression. EGFR expression had a linear relationship with the number of metastatic lymph nodes [18]. Study authors suggested that expression of VEGF and EGFR were independent prognostic indicators of worse outcomes for GC patients. In the study performed by Rao et al., serum VEGF levels were elevated in comparison to healthy controls and significantly increased in patients with advanced tumor TNM stages (*p* < 0.05). They concluded that VEGF might be an effective indicator for the evaluation of the prognosis of GC patients [49]. Vidal et al. assessed VEGF levels in GC patients. Elevated VEGF concentrations were correlated with advanced TNM stage, lymph nodes invasion, a lower probability of recurrence-free status and shorter disease-specific survival of GC patients [50].

### 3.3. Specific Proteins—Interleukin 6 (IL-6) and C-Reactive Protein (CRP)

Interleukin 6 is a proinflammatory cytokine that stimulates the production of other acute-phase proteins, such as CRP, α1-antichymotrypsin, fibrinogen, SAA and haptoglobin. This cytokine is involved in the immune response, inflammation processes and hematopoiesis, as well as in many pathological conditions, including malignancies such as breast cancer, lung cancer, pancreatic cancer and GC [51]. It was proved that IL-6 stimulates the synthesis of C-reactive protein (CRP). CRP is an acute-phase protein produced by hepatocytes in the liver. The serum concentration of CRP in healthy patients is below 10 mg/L; however, its level may be elevated during many pathological conditions, such as bacterial infections, as well as malignant diseases, including GC [52]. Matsuo et al. assessed IL-6 expression in nine GC cell lines and nine colorectal cancer cell lines. Two GC and one colorectal cancer cell lines expressed IL-6. The level of IL-6 secretion was higher in GC cell lines than in colorectal cancer cell lines, and this difference was statistically significant [19]. The study of Wang et al. has indicated that the expression of IL-6 in GC patients was higher than in the control group, and this difference was statistically significant [20]. In addition, Chang et al. revealed that elevated CRP levels were observed in 38% of GC patients and in 4.9% of the patients from the control group, and these differences were statistically significant [53]. A higher concentration of serum CRP was also associated with larger tumor size, the presence of lymph nodes and distant metastases, as well as more advanced stages of GC and worse survival rates of patients [53]. In our previous research, we assessed serum IL-6, CEA and CA19-9 levels in GC patients and compared them with healthy controls [21]. Moreover, our data revealed that serum CRP concentration correlated with the presence of lymph node and distant metastases, advanced cancer stage and gastric wall invasion, while IL-6, CEA and CA 19-9 levels correlated with nodal involvement [21]. Additionally, the diagnostic sensitivity of IL-6 was higher than for CRP, CA 19-9 and CEA and increased in combined assessment with CRP or CEA [21]. Our study revealed better usefulness of inflammation proteins IL-6 and CRP than classical GC markers such as CEA and CA 19-9 in the diagnosis and progression of this malignancy [21].

### 3.4. Matrix Metalloproteinases (MMPs) and Their Tissue Inhibitors (TIMPs)

Matrix metalloproteinases (MMPs), also known as matrix metallopeptidases, are calcium-dependent zinc-containing endopeptidases. They are produced by leukocytes, macrophages, endothelial cells, fibroblasts and tumor cells [54]. Tissue inhibitors of metalloproteinases (TIMPs) are specific endogenous protease inhibitors that inhibit MMPs and regulate the process of their activation [55]. There are four types of TIMPs: TIMP-1, TIMP-2, TIMP-3 and TIMP-4. Several studies suggested the usefulness of selected MMPs and their tissue inhibitors as novel biomarkers of malignancies, including GC. Mucosal MMP-1 mRNA levels were higher in GC with active *H. pylori* infection than in GC without active *H. pylori* infection [24]. Laitinen et al. revealed that serum MMP-9 and TIMP-1 might be used as prognostic biomarkers of GC. In addition, a worse prognosis was observed among GC patients with high serum levels of TIMP-1 [56]. Kasurinen et al. assessed that patients with elevated serum MMP-14 levels had a 5-year disease-specific survival of 22.1%, while GC patients with lower concentrations of MMP-14 had a 5-year specific survival of 49.2%, and this difference was statistically significant (*p* = 0.01) [25]. The authors concluded that serum MMP-14 was a marker of poor prognosis as well as an indicator of the presence of distant metastases [25]. In our previous study, we assessed the diagnostic significance of gelatinases such as MMP-9 and MMP-2 as well as their tissue inhibitors (TIMP-1 and TIMP-2) in GC patients. We indicated that the diagnostic sensitivity of MMP-2 and TIMP-2 was higher in GC and inflammatory cells compared to normal tissue, whereas serum levels of these proteins were statistically lower in GC patients than in healthy subjects. In addition, our paper demonstrated that there was a significant positive correlation between TIMP-2 immunoreactivity in inflammatory cells and the presence of lymph node metastasis. In addition, we revealed that the area under the ROC curve (AUC) for TIMP-2 was higher than MMP-2, while serum MMP-2 was proved to be an independent prognostic factor of GC patients’ survival. Thus, our findings suggest that TIMP-2 might be a predictor of GC progression, especially for nodal involvement, while serum MMP-2 can be used as an independent prognostic factor of patient survival [22]. Previously we evaluated that plasma and serum MMP-9, its tissue inhibitor TIMP-1 and classical tumor marker (CEA) levels were significantly higher in GC patients compared with healthy controls. Moreover, diagnostic criteria such as AUC, diagnostic sensitivity and accuracy of plasma TIMP-1 were higher than those for MMP-9 and CEA, whereas an increased plasma TIMP-1 level was a significant independent prognostic factor for the survival of GC patients. Based on our findings, we suggest that plasma TIMP-1 is a better biomarker than serum TIMP-1 and might be useful for the diagnosis of GC and the prognosis of patient survival [23].

### 3.5. A Disintegrin and Metalloproteinase with Thrombospondin Motifs (ADAMTS) 

A disintegrin and metalloproteinase with thrombospondin motifs (ADAMTS) is a family of multidomain extracellular protease enzymes. Some clinical investigations suggest that ADAMTS12 is related to oncogenesis. In cancer-related processes, ADAMTS12 shows dual effects, being pro and anti-tumor in a proteolytic or non-proteolytic manner. Hou et al. indicated that ADAMTS12 was upregulated in GC patients and predicted a worse overall survival rate in GC patients. In addition, based on AUC analysis, the authors demonstrated that ADAMTS12 had a certain predictive value for the diagnosis of GC [26]. In the study of Jiang et al., ADAMTS2 expression was elevated in the cytoplasm of gastric tumor cells and fibroblast cells. Upregulated expression of ADAMTS2 was correlated with the type of Lauren classification and TNM stage of GC. Lower expression of ADAMTS2 was a good prognostic factor of GC patients’ survival. Additionally, multivariate analysis indicated that ADAMTS2 expression was an independent prognostic factor, and this protein may be used as a potential biomarker for GC prognosis [27]. The study of Chen et al. assessed the downregulation of ADAMTS8 mRNA expression in GC cell lines and tissues. There was a significant correlation between ADAMTS8 expression and higher depth of tumor invasion, and the presence of lymph node metastasis. ADAMTS mRNA was also significantly lower in methylated primary gastric tumors than in nontumor tissues. Methylation of the ADAMTS8 gene was statistically higher in primary gastric tumors than in nontumor tissues. There was a significant association between ADAMTS18 methylation and lymph node metastasis [28]. The authors suggest that selected ADAMTS might have promising usefulness as novel biomarkers of GC. 

### 3.6. DNA-Based Biomarkers

DNA methylation plays an important role in GC development. It is suggested that methylated DNA can be served as a biomarker in the plasma, serum or in gastric washes of a cancer patient, including GC. Altered DNA methylation was also observed in patients with *H. pylori*-infected gastric mucosa. Frequently methylated genes in primary GC are: *p16*, *RUNX3*, *CDH1*, *APC*, *DAPK*, *GSTP1*, *MLH1*, *LOX*, *FLNc*, *HRASL*, *HAND*, *THBD*, *F2R*, *NT5E*, *GREM*, *ZNF177*, *CLDN3*, *PAX6*, *CTSL*, *ALX4*, *TMEFF2*, *CHCHD10*, *IGFBP3*, *NPR1*, *CHFR ADAMTS9*, *FOXD3* and *PAX5* [57,58,59,60,61,62,63]. In the Li et al. study, *PAX5* hypermethylation was detected in 77% of primary GC tissues compared to 10.5% of normal gastric tissues (*p* < 0.0001) [63]. Moreover, GC patients with *PAX5* methylation had worse survival compared with unmethylated cases. It was also concluded that *PAX5* might be a suppressor in GC, and the detection of methylated *PAX5* can be used as an independent GC prognostic factor [63]. In addition, some authors revealed that the presence of *H. pylori* infection, which is a well-established risk factor of GC, causes epigenetic deregulation of *FOXD3* to promote gastric carcinogenesis. *FOXD3* hypermethylation in GC was significantly elevated compared with adjacent preneoplastic tissues. Patients with higher *FOXD3* methylation levels survived for significantly shorter time than the patients with lower *FXDP3* methylation levels. Circulating cell-free DNA (cfDNA) is cell-free extracellular DNA from both normal and cancer cells. Some investigations have shown the presence of circulating DNA containing tumor-specific genetic information in the peripheral blood of patients with cancer. The most studied subject connected with cfDNA is ctDNA originating from primary tumors, metastases or circulating tumor cells [64]. Park et al. assessed circulating DNA levels in GC patients and the control group. These results revealed that the plasma cfDNA concentration was significantly higher in patients with GC than that in healthy patients. The authors suggested that plasma is a better source of cfDNA as a biomarker than serum. The optimal cut-off value of plasma cfDNA concentration for discriminating between GC and healthy patients was 32.3 ng/mL. The diagnostic sensitivity of this biomarker was estimated at 75%, while the diagnostic specificity was 63% in GC patients. In addition, the AUC for cfDNA was 0.784 [65]. The presented results confirm the potential role of DNA-based biomarkers in the diagnosis of GC patients. 

### 3.7. RNA-Based Biomarkers

A growing body of evidence indicates the importance of novel RNA-based biomarkers in GC. MicroRNAs (miRNAs) consist of 20–25 nucleotides, and they are short non-coding RNAs. Their role may be involved in carcinogenesis [66]. The expression of microRNA-34a was lower in GC tissues, metastatic GC tissues and in more advanced stages of the cancer than in the control group. Moreover, microRNA-34a expression was elevated in GC tissues without metastases. Additionally, serum microRNA-34a level in GC patients was also higher than in healthy controls and correlated with better prognosis of GC patients’ survival [67]. Recent studies established some promising biomarkers which can diagnose GC using non-invasive methods in urine samples. These biomarkers are urinary microRNAs such as miR-6807-5p and miR-6856-5p. It was demonstrated that urinary micro-RNAs had higher expression in GC patients than in the healthy control group. Serum levels of miR6807-5p and miR-6856-5P were also higher in stage I of the GC than in the control group, and this difference was statistically significant [68]. Moreover, the authors suggest that this panel of urinary microRNA biomarkers may be used in the diagnosis of GC patients in the early stage of the disease. In the presented study, researchers also compared the levels of assessed parameters in patients before and after the tumor resection and revealed that the urinary levels of these miRNAs decreased to the undetectable level in all cases, which may suggest the promising role of RNA-based biomarkers in the diagnosis of patients with this malignancy [68].

### 3.8. C-MET (Tyrosine-Protein Kinase Met)

Some clinical investigations have suggested the potential role of c-MET (tyrosine-protein kinase Met), also known as hepatocyte growth factor receptor (HGFR), in GC pathogenesis [69,70,71]. It is a protein encoded by the *MET* gene. Activation of MET phosphorylates transduction cascade may promote tumor growth, angiogenesis, migration of the cells and metastasis [67]. Zhang et al. indicated that the expression of c-MET was higher in GC tissues than in paracancerous tissues [72]. Moreover, there was a statistically significant difference between the c-MET expression and clinicopathological characteristics of GC. Elevated expression of c-MET correlated with higher M-stage from TNM staging system (M—the presence of distant metastasis). Overexpression of c-MET was associated with poor overall survival of GC patients. In addition, multivariate Cox regression analysis showed that c-Met might be used as an independent risk factor for 5 years of survival after surgery [73]. In a study performed by Tsujio, HER2-positive GC patients with the c-MET positive expression had worse overall survival than subjects with c-MET negative expression. c-MET positive expression was also more often observed with patients in N1 lymph node metastasis than without nodal involvement (N0 subgroup) [74]. Moreover, c-MET may also be an important biomarker in the treatment process of GC patients. Yashiro et al. indicated that the inhibition of c-MET increased the chemosensitivity of cancer stem cells to the irinotecan in GC, which suggested the role of this biomarker in the monitoring of treatment [70].

### 3.9. Other Molecular Biomarkers of GC 

The molecular characterization of GC has been under excessive investigation; therefore, the identification of novel biomarkers and therapeutic targets is sorely needed. 

Fibroblast growth factor receptors (FGFRs) are a family of tyrosine kinase receptors (RTKs). Their signaling is important in processes such as the proliferation or invasion of tumor cells [75]. There is a potential GC therapeutic target connected with this factor: bemarituzumab. It is an afucosylated, humanized IgG1 anti-fibroblast growth factor receptor 2 isoform IIb (FGFR2b) monoclonal antibody. There was a two-phase study performed on patients with FGFR2b-selected gastric or gastro-esophageal junction adenocarcinoma. The study revealed that bemarituzumab has promising clinical efficiency [76].

Epstein-Barr virus (EBV) is a member of the herpes virus family. It was originally identified in a human Burkitt lymphoma cell line. EBV infects more than 90% of the population, and most of those infections are asymptomatic. However, in individuals, it increases the risk of Burkitt lymphoma, Hodgkin lymphoma, nasopharyngeal carcinoma and gastric adenocarcinoma [77]. Epstein–Barr-virus-associated gastric cancer (EBVaGC) occurs in 2–20% of GC cases. It occurs more often among males than females and among Caucasians than Asians. It is a distinct molecular GC subtype with a better prognosis and fewer lymph node metastases [78,79]. Bai et al. revealed that in GC patients with DNA mismatch repair proficiency (pMMR), EBV status was concluded to be an independent predictive factor for overall survival and survival without progression [80].

Cytokeratin-19 (CK19) is expressed by cancer cells and may be used as a marker of metastases in GC [81]. Kutun et al. assessed the patients with resectable GC, unresectable GC and the control group and concluded that expression of both CEA and CK-19 in the peripheral blood of GC patients are strong major vascular invasion (MVI) predictors and worse survival [82]. In addition, microsatellite instability (MSI) and Epstein-Barr virus (EBV) status in GC were also proved to be important biomarkers regarding overall survival and progression-free survival, as well as response to perioperative chemotherapy [83]. The authors indicated that female patients with MSI-high malignancy had significantly better overall survival than females with microsatellite stable (MSS) tumors when submitted to response to perioperative chemotherapy, whereas opposite findings were revealed in male patients. The presented results support research concerning the personalized treatment of GC, considering both patients’ and disease characteristics [83].

PD1 (programmed cell death 1) receptor and its ligand PD-L1 are involved in immunomodulation. PD1/PD-L1-Immunotherapy is approved for GC. A high PD-L1/PD1 expression was associated with a better outcome for the patient and was an independent factor in overall and tumor-specific survival prognosis [84,85].

The role of neutrophic thyrosine receptor kinase (*NTRK*) in GC was assessed in several studies, but it was suggested that the role is marginal as the occurrence of *NTRK* fusion in GC is very rare [86,87,88]. In spite of this, there is entrectinib, which has anti-cancer activity in GC cells with *NTRK* overexpression. It stops angiogenesis and cancer progression by apoptosis induction [89]. It may be used in the treatment of solid, advanced or metastatic tumors, such as GC. It was approved by the FDA in 2019 for adult patients with ROS1-positive metastatic non-small cell lung cancer [90].

In GC treatment, a positive association between tumor mutational burden (TMB) and clinical outcomes in GC patients with pembrolizumab was observed [91]. Lee et al. assessed the association between TMB status and first-line pembrolizumab with chemotherapy treatment outcomes in addition to chemotherapy in KEYNOTE-062. The study revealed the clinical efficiency of first-line with pembrolizumab-based therapy for patients with advanced gastric/gastroesophageal junction adenocarcinoma [92]. 

## 4. Methods for Evaluating GC Biomarkers 

In routine practice, serum concentrations of classical tumor markers (for example, CEA) are measured using the standard, low-cost chemiluminescent method in routine analyzers with ready-to-use calibrators and controls. However, the diagnostic sensitivity and specificity of classical tumor markers are insufficient to use these biomarkers in the early detection of malignancy. There is also a need for further analyses, considering the relationship between the technical difficulties of tumor marker detection methods and their cost-effectiveness with the diagnostic accuracy of GC biomarkers. We summed up sources and methods of assessed biomarkers detection in a flowchart to help to understand their clinical applicability (Figure 2). The most common diagnostic methods for novel GC biomarkers that were used in presented studies differ significantly in terms of technical complexity and costs. Collecting the tumor specimens for immunohistochemistry (IHC) is a more invasive method than the assessment of the biomarkers in serum or in plasma. That assessment may be used in immunoenzyme techniques, such as ELISA or multiplex technology. IHC is also a more complicated and time-consuming technique in comparison to faster, cheaper and simpler immunoenzyme methods. Enzyme-linked immunosorbent assay (ELISA) is a highly sensitive method that is widely used in research. It involves many steps of analysis that take even 6 h to complete the measurement of a single biomarker. On the other hand, multiplex technology is able to measure up to 500 protein targets combined from a single complex biological sample and can also be used to assess RNA and protein targets as well as gene expression. 

Culture of tissue or fluid remains the current standard of care for cancer diagnosis, including GC. However, molecular techniques seem to be the future of early diagnosis of malignant diseases. The costs of applying diagnostic techniques based on polymerase chain reaction (PCR) are known to be higher than ELISA or multiplex technology, but these methods improve the understanding of the nature and biology of a tumor [93]. PCR is a molecular biology technique used to amplify DNA. It is detected as the reaction progresses in real-time, and reverse transcription-PCR (RT-PCR) allows for quantification of the levels of messenger or ribosomal RNA and evaluates the level of protein synthetic activity. Molecular diagnostic techniques are widely applied, powerful and sensitive. They are used to identify biomarkers in the genome and proteome [94]. An assessment of a biomarker level in body fluids in combination with molecular tests may, in the future, be the best method in the cancer diagnostic process, including GC diagnostics. 

## 5. Conclusions

Biomarkers for early detection of CRC are important to improve the management of patients with GC. The diagnostic specificity and sensitivity of well-established biochemical markers are limited. Therefore, there is a need for new non-invasive, safe, easily measurable and low-cost methods in the cancer diagnostic process. Our review presents the potential usefulness of novel biomarkers and compares their significance with the well-established tumor markers for this malignancy, including CA 72-4, CEA and CA 19-9.

The present paper indicates that selected chemokines and their receptors, VEGF and EGFR, DNA- and RNA-based biomarkers, and MET, IL-6 and CRP, as well as MMPs and their tissue inhibitors, are promising biomarkers in the diagnosis and progression of GC, while selected ADAMs and TIMPs might be used as potential prognostic factors of a GC patient survival.

Presented findings suggest that novel biomarkers may help in establishing an accurate diagnosis and in the treatment of cancer, in choosing medications for a patient or in predicting the drug response. They can also be used in the follow-up of cancer survivors. There is a need for more studies on larger groups connected with this subject to take a new biomarker into routine use; however, there are some promising candidates to improve the GC diagnostic and therapeutic process. 

## Figures and Tables

**Figure 1 ijms-24-08833-f001:**
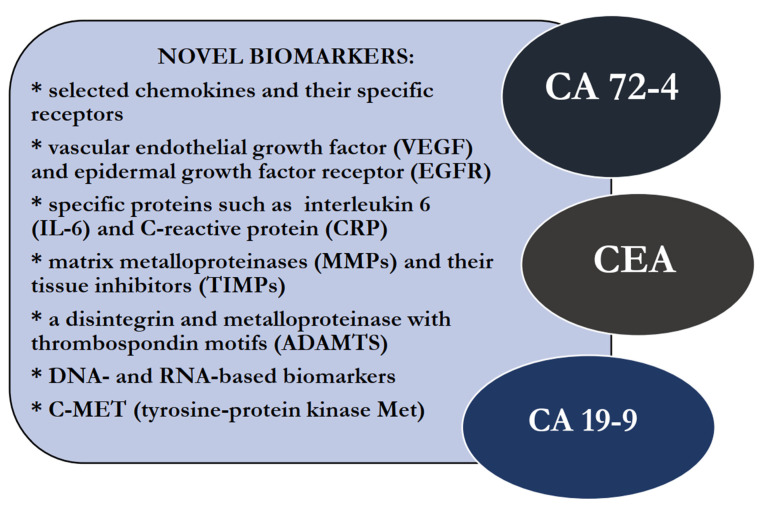
Potential, novel biomarkers and well-established tumor markers for gastric cancer (GC).

**Figure 2 ijms-24-08833-f002:**
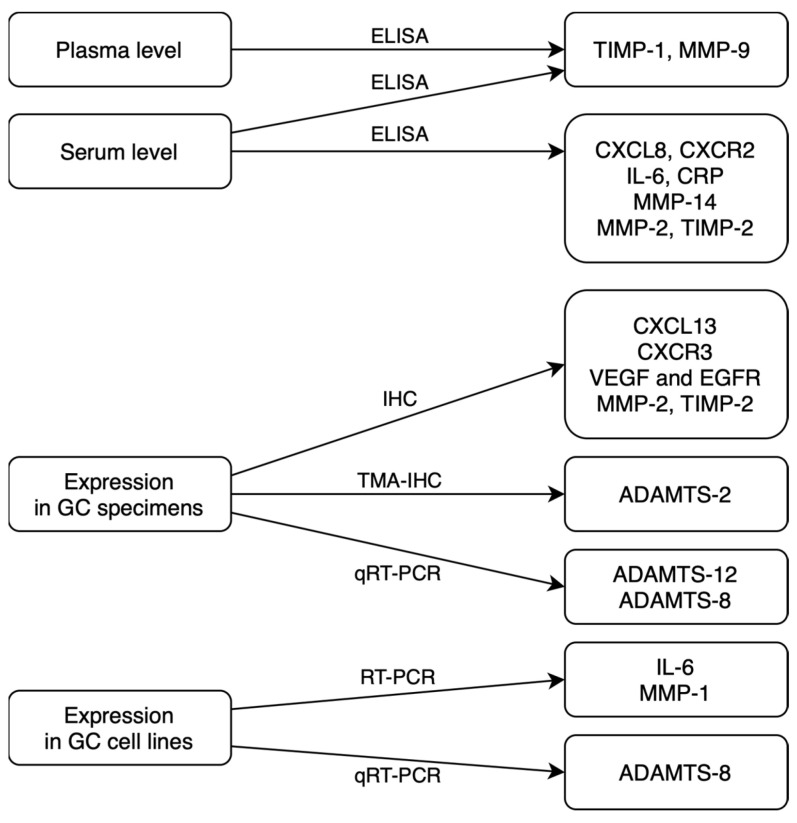
Sources and methods of potential GC biomarkers detection (ELISA—enzyme-linked immunosorbent assay; IHC—immunohistochemistry; TMA-IHC—tissue microarray immunohistochemistry; qRT-PCR—quantitative reverse transcription polymerase chain reaction) [15,16,17,18,19,21,22,23,24,25,26,27,28].

**Table 1 ijms-24-08833-t001:** Significance of novel biomarkers in gastric cancer (GC).

Potential Biomarker	Source	Results	References
CXCL8 and CXCR2	Serum level	✓ Higher levels of CXCL8 and CXCR2 in GC patients than in the healthy control group. ✓ Potential significance in the diagnosis of GC patients.	[15]
CXCL13	Expression	✓ Elevated expression of CXCL13 is associated with a larger tumor diameter and a shorter overall survival rate. ✓ Low expression of CXCL13 is associated with a longer survival rate, especially in the group of patients who received adjuvant chemotherapy.	[16]
CXCR3	Expression	✓ Higher expression of CXCR3 in GC tissues than in precancerous tissues. ✓ Overexpression of CXCR3 correlated with decreased M2 macrophage infiltration. ✓ Lower CXCR3 expression correlated with worse differentiation, more advanced GC stage and higher depth of invasion.	[17]
VEGF and EGFR	Expression	✓ Elevated VEGF and EGFR expressions associated with nodal invasion and tumor progression.	[18]
IL-6 and CRP	Expression	✓ Higher IL-6 expression in GC cell lines than in colorectal cancer cell lines.	[19]
		✓ Higher IL-6 expression in GC patients than in the control group.	[20]
	Serum level	✓ Potential significance of IL-6 and CRP in the diagnosis of GC patients.	[21]
TIMPs	Serum level	✓ TIMP-2 is useful in predicting GC tumor progression, especially for nodal involvement.	[22]
	Serum and plasma level	✓ Plasma TIMP-1 is a better GC biomarker than serum TIMP-1. ✓ Plasma TIMP-1 may be useful in the prognosis of a GC patient's survival.	[23]
MMPSs	Mucosal mRNA expression	✓ MMP-1 mRNA levels are higher in GC with active *H. pylori* infections than in GC without *H. pylori* infections.	[24]
	Serum level	✓ GC patients with elevated serum MMP-14 levels had a 5-year disease-specific survival of 22.1% ✓ GC patients with lower concentrations of MMP-14 had a 5-year specific survival of 49.2%.	[25]
		✓ MMP-2 may be used as an independent prognostic factor of a GC patient's survival.	[22]
	Serum and plasma level	✓ Plasma and serum levels of MMP-9 were significantly higher in GC patients than in the control group.	[23]
ADAMTS	Expression	✓ ADAMTS-12 upregulation in GC patients and prediction of worse survival rate of GC patients.	[26]
	Expression	✓ Lower expression of ADAMTS-2 was a good prognostic factor of GC patient survival.	[27]
	Expression	Downregulation of ADAMTS-8 expression in GC cell lines and tissues.	[28]

## Data Availability

Not applicable.

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
