# Peer review of "The Comprehensive Analysis of Specific Proteins as Novel Biomarkers Involved in the Diagnosis and Progression of Gastric Cancer"

_ijms, 2023, doi:10.3390/ijms24108833_

Round 1

Reviewer 1 Report

I think it is a good review. It's shown a very good spectrum of possibilities of biomarkers and their possible utility both in the spectrum of diagnosis and in the estimation of probability of extended disease and the prognosis patients survival. Perhaps could be improved in the discussion to make more emphasis on those that for the authors are outlined as more useful in clinical practice in the neaxt time. Along with this, it would be desirable for them to make comments regarding the differences in technical difficulties, difficulties in technique standardizing, and costs so that the clinical reader can get an idea of which ones can be outlined to be included in their local reality.

Author Response

DETAILED ANSWERS TO THE REVIEWERS’ COMMENTS

Reviewer 1

I think it is a good review. It's shown a very good spectrum of possibilities of biomarkers and their possible utility both in the spectrum of diagnosis and in the estimation of probability of extended disease and the prognosis patients survival. Perhaps could be improved in the discussion to make more emphasis on those that for the authors are outlined as more useful in clinical practice in the neaxt time. Along with this, it would be desirable for them to make comments regarding the differences in technical difficulties, difficulties in technique standardizing, and costs so that the clinical reader can get an idea of which ones can be outlined to be included in their local reality.

Authors' Responses to Reviewer's Comments

We are very grateful for your kind review and suggestions. New chapter concerning technical difficulties of presented methods for biomarkers analysis has been added in the revised manuscript, according to Reviewer 1 suggestion (pages 12-13, lines 388-420). Moreover, new references have also been added in the new version of manuscript, as it was recommended (page 19, lines 702-706). In addition, a flowchart including sources and methods of GC biomarkers assessment has been added in the revised  paper to help to understand clinical applicability (page 13, lines 421-426).

Reviewer 2 Report

The reviewed paper addresses the topical issue of finding new biomarkers for gastric cancer. The manuscript is arranged in a logical manner. Nonetheless, from my perspective, there is a lack of data to fully comprehend markers used in gastric cancer:

1. there are studies on FGFR in gastric cancer and the potential drug bemarituzumab

2. Determining HER2 status is only part of the process, as assessing biomarkers like MSI status, MMR, PD-L1, TMB-H, and NTRK expression is also essential when looking into potential treatment.

3. EBV status, and CK-19 mRNA in peritoneal lavage are also considered potential biomarkers

Author Response

DETAILED ANSWERS TO THE REVIEWERS’ COMMENTS

Reviewer 2

The reviewed paper addresses the topical issue of finding new biomarkers for gastric cancer. The manuscript is arranged in a logical manner. Nonetheless, from my perspective, there is a lack of data to fully comprehend markers used in gastric cancer:

  1. there are studies on FGFR in gastric cancer and the potential drug bemarituzumab

Authors' Responses to Reviewer's Comments

We are very grateful for your kind review and suggestions. New chapter concerning the significance of other molecular biomarkers for GC such as FGFR in GC has been added in the revised version of manuscript, as it was recommended (page 11, lines 340-346). Moreover, new references have also been added in the new version of the manuscript (page 18, lines 648-654).

  1. Determining HER2 status is only part of the process, as assessing biomarkers like MSI status, MMR, PD-L1, TMB-H, and NTRK expression is also essential when looking into potential treatment.

Authors' Responses to Reviewer's Comments

We are very grateful for your kind review and suggestions. New chapter concerning the significance of other molecular biomarkers for GC such as MMR, PD-L1, TMB-H, and NTRK in GC has been added in the revised version of manuscript, as it was recommended (pages 12, lines 369-386). Moreover, new references have also been added in the new version of manuscript (page 19, lines 671-701).

  1. EBV status, and CK-19 mRNA in peritoneal lavage are also considered potential biomarkers

Authors' Responses to Reviewer's Comments

We are very grateful for your kind review and suggestions. New chapter concerning the significance of other molecular biomarkers for GC such as EBV status, and CK-19 mRNA in GC has been added in the revised version of manuscript, as it was recommended (pages 11-12, lines 347-368). In addition, new references have also been added in the new version of manuscript (pages 18-19, lines 655-670).

Reviewer 3 Report

In this well written review article authors have provided information of biomarkers that can be used to detect the gastric cancer.  

In future studies......Correlating these specific proteins with patient metabolism may also produce great potential to cancer diagnosis.

1. While authors have given the details information, it will be more better if there is any flow chart that is representing the soucrce (sample i.e serum, tissue) of protein and its detection methods.

2. There could be small introduction of current methodology update being used for the small protein detection.

3. Reference should be added 

     line 49-52 i.e. for MRI

Author Response

DETAILED ANSWERS TO THE REVIEWERS’ COMMENTS

Reviewer 3

In this well written review article authors have provided information of biomarkers that can be used to detect the gastric cancer.  In future studies......Correlating these specific proteins with patient metabolism may also produce great potential to cancer diagnosis.

  1. While authors have given the details information, it will be more better if there is any flow chart that is representing the soucrce (sample i.e serum, tissue) of protein and its detection methods. 

Authors' Responses to Reviewer's Comments

Thank you very much for your kind review and very useful remarks. A flow chart that is representing the source of protein and its detection methods has been added in the revised version of manuscript, according to Reviewer 3 suggestion (page 13, lines 421-426).

  1. There could be small introduction of current methodology update being used for the small protein detection.

Authors' Responses to Reviewer's Comments

New chapter concerning current methodology being used for the biomarkers detection has been presented in the new version of paper (pages 12-13, lines 388-420), thus new references have also been added in the revised manuscript, as it was recommended (page 19, lines 702-706).

  1. Reference should be added - line 49-52 i.e. for MRI

Authors' Responses to Reviewer's Comments

We replaced mistakenly used reference in the fragment related to MRI and other diagnostic methods with an appropriate article. The reference no 8 has been corrected in the revised version of manuscript, according to the Reviewer 3 suggestion (page 14, lines 464-466).

Round 2

Reviewer 2 Report

Accept in present form

Reviewer 3 Report

Authors have done the changes as suggested. I recommend this review for publication without any changes.